# Drug Resistance Patterns of Commonly Used Antibiotics for the Treatment of *Helicobacter pylori* Infection among South Asian Countries: A Systematic Review and Meta-Analysis

**DOI:** 10.3390/tropicalmed8030172

**Published:** 2023-03-14

**Authors:** Abhigan Babu Shrestha, Pashupati Pokharel, Unnat Hamal Sapkota, Sajina Shrestha, Shueb A. Mohamed, Surakshya Khanal, Saroj Kumar Jha, Aroop Mohanty, Bijaya Kumar Padhi, Ankush Asija, Yub Raj Sedhai, Rishikesh Rijal, Karan Singh, Vijay Kumar Chattu, Alfonso J. Rodriguez-Morales, Joshuan J. Barboza, Ranjit Sah

**Affiliations:** 1M Abdur Rahim Medical College, Dinajpur 5200, Bangladesh; 2Department of Medicine, Maharajgunj Medical Campus, Institute of Medicine, Tribhuvan University, Kathmandu 1524, Nepal; 3Department of Internal Medicine, KIST Medical College, Imadol, Patan 284128, Nepal; 4School of Medicine, Alexandria University, Alexandria 21568, Egypt; 5Department of Microbiology, All India Institute of Medical Sciences, Gorakhpur 273008, India; 6Department of Community Medicine and School of Public Health, Postgraduate Institute of Medical Education and Research, Chandigarh 160012, India; 7School of Medicine, West Virginia University, Morgantown, WV 26506, USA; 8Division of Pulmonary Disease and Critical Care Medicine, University of Kentucky College of Medicine, Bowling Green, KY 42101, USA; 9Division of infectious Diseases, University of Louisville, Louisville, KY 40208, USA; 10Department of Occupational Science and Occupational Therapy, Temerty Faculty of Medicine, University of Toronto, Toronto, ON M5R 0A3, Canada; 11Center for Transdisciplinary Research, Saveetha Dental College, Saveetha Institute of Medical and Technical Sciences, Saveetha University, Chennai 600077, India; 12Department of Community Medicine, Faculty of Medicine, Datta Meghe Institute of Medical Sciences, Wardha 442107, India; 13Grupo de Investigación Biomedicina, Faculty of Medicine, Fundación Universitaria Autónoma de las Américas, Pereira 660003, Colombia; 14Master’s Program in Clinical Epidemiology and Biostatistics, Universidad Cientifica del Sur, Lima 15846, Peru; 15Gilbert and Rose-Marie Chagoury School of Medicine, Lebanese American University, Beirut 1101, Lebanon; 16Escuela de Medicina, Universidad Cesar Vallejo, Trujillo 13007, Peru; 17Department of Microbiology, Dr. D. Y. Patil Medical College, Hospital and Research Centre, Dr. D. Y. Patil Vidyapeeth, Pune 411018, India; 18Department of Public Health Dentistry, Dr. D.Y. Patil Dental College and Hospital, Dr. D.Y. Patil Vidyapeeth, Pune 411018, India

**Keywords:** antibiotic resistance, *Helicobacter pylori*, South Asia, meta-analysis, metronidazole, clarithromycin, tetracycline, fluroquinolones, amoxicillin

## Abstract

Background: In South Asia, resistance to commonly used antibiotics for the treatment of *Helicobacter pylori* infection is increasing. Despite this, accurate estimates of overall antibiotic resistance are missing. Thus, this review aims to analyze the resistance rates of commonly used antibiotics for the treatment of *H. pylori* in South Asia. Methods: The systematic review and meta-analysis was conducted in accordance with the Preferred Reporting Items for Systematic Reviews and Meta-Analysis statement. We searched five medical databases for relevant studies from inception to September 2022. A random effect model with a 95% confidence interval (CI) was used to calculate the pooled prevalence of antibiotic resistance. Results: This systematic review and meta-analysis included 23 articles, 6357 patients, 3294 *Helicobacter pylori* isolates, and 2192 samples for antibiotic resistance. The prevalences of antibiotic resistance to common antibiotics were clarithromycin: 27% (95%CI: 0.17–0.38), metronidazole: 69% (95%CI: 0.62–0.76), tetracycline: 16% (95%CI: 0.06–0.25), amoxicillin: 23% (95%CI: 0.15–0.30), ciprofloxacin: 12% (95%CI: 0.04–0.23), levofloxacin: 34% (95%CI: 0.22–0.47), and furazolidone: 14% (95%CI: 0.06–0.22). Subgroup analysis showed antibiotic resistances were more prevalent in Pakistan, India, and Bangladesh. Furthermore, a ten-year trend analysis showed the increasing resistance prevalence for clarithromycin (21% to 30%), ciprofloxacin (3% to 16%), and tetracycline (5% to 20%) from 2003 to 2022. Conclusion: This meta-analysis showed a high prevalence of resistance among the commonly used antibiotics for *H. pylori* in South Asian countries. Furthermore, antibiotic resistance has been increasing over the time of 20 years. In order to tackle this situation, a robust surveillance system, and strict adherence to antibiotic stewardship are required.

## 1. Introduction

*Helicobacter pylori* is a Gram-negative, spiral-shaped, microaerophilic flagellated bacillus colonizing the human stomach [1]. *H. pylori* infection has a wide range of presentations ranging from asymptomatic to complicated and even life-threatening conditions such as peptic ulcer, superficial and chronic gastritis, to even as a prequel of gastric adenocarcinoma [2]. Accordingly, the International Agency for Research on Cancer (IARC) has classified *H. pylori* as a class I carcinogen for gastric carcinoma [3]. 

A recent meta-analysis has estimated the prevalence of *H. pylori* infection to be 57.7% in South Asia, which is higher than the prevalence in Asia (44.7%) [4]. A study showed that chronic *H. pylori*-infected people were predisposed three times to non-cardia gastric cancer (odds ratio 3.0; 95% confidence interval: 2.3–3.8) [5]. With the high prevalence of infection and high incidence of gastric carcinoma, treatment strategies for the eradication of *H. pylori* infection should be more robust in South Asia.

Treatment guidelines for *H pylori* infection are guided by the Maastricht/Florence consensus report, the latest of which is the sixth version published in 2022. According to consensus, the first-line recommended treatment in areas of high (>15%) or unknown clarithromycin resistance is bismuth-containing quadruple therapy. If this is not available, non-bismuth concomitant quadruple therapy may be considered. However, in areas of low clarithromycin resistance, bismuth quadruple therapy or clarithromycin-containing triple therapy is recommended as first-line empirical treatment, if proven effective locally. The duration of treatment is usually 14 days unless a 10-day effective regimen is available [6]. 

However, with the ubiquitous and burgeoning use of antibiotics, the efficacy of this standard triple regimen has decreased over the past few years with dramatically increasing antibiotic resistance in *H. pylori* [7]. The prevalence of antibiotic resistance in *H. pylori* appears to be regionally variable and antibiotic specific, although overall antibiotic resistance appears to be markedly increasing over time, with a concordant decline in the antimicrobial eradication rate of *H. pylori* globally [8]. The fact of greater interest is that the available alternative therapies including quadruple, sequential, concomitant, and levofloxacin-containing triple regimens have varied considerably in treating infection [8,9]. Even if the selection of treatment based on local resistance patterns has been strongly recommended by international consensus reports, *H. pylori* antibiotic sensitivity testing is rarely performed, especially among the poor and underdeveloped or developing countries, thus further contributing to drug resistance [10,11].

Globally, resistance rates are variable for the antibiotics used for the treatment of *H. pylori* infection, ranging from 15% to 50% [8]. However, the resistance patterns are considerably high among Asian countries [12]. South Asia, comprising eight countries and one-fourth of the global population, has a high prevalence of antibiotic resistance due to a high self-medication rate, poor antibiotic stewardship plans, and changing geopolitical landscape [13]. This has led to high antibiotic resistance in *H. pylori* of up to 98% [14,15]. A proper understanding of the antibiotic resistance pattern in *H. pylori* through timely and systematic analysis of the available regional data thus is a promising strategy to eliminate *H. pylori*-borne infection and its long-term consequences. 

Despite the high prevalence of *H. pylori* infection and antibiotic resistance in South Asia, limited studies have been performed regarding antibiotic resistance patterns, and the studies performed were not systematically reviewed, and, therefore, were prone to selection bias. Therefore, the primary objective of this study was to evaluate the prevalence of the antibiotic resistance of *H. pylori* in the South Asia region. 

## 2. Materials and Methods

### 2.1. Study Protocol

We piloted this systematic review and meta-analysis in accordance with the Preferred Reporting Items for Systematic Reviews and Meta-Analyses (PRISMA) guidelines [16]. The protocol for this study with predefined methodology was registered on PROSPERO (CRD42021264656). The PRISMA checklist is detailed in the Appendix A.

### 2.2. Search Strategy 

For the relevant articles, the databases of PubMed, Embase, Cochrane Library, Google Scholar and the Web of Science from the time of inception to September, 2022 were searched systematically using appropriate Boolean operators, generic terms, keywords, and medical subject headings (MeSH): “*H. pylori*”, “*Helicobacter pylori*”, “antibiotic”, “antibacterial”, “antimicrobial”, “resistance”, “India”, “Bhutan”, “Maldives”, “Nepal”, “Bangladesh”, “Sri Lanka”, “Afghanistan”, and “Pakistan”. The search was restricted to include cross-sectional studies only, to minimize the risk of heterogeneity arising from differences in the study design. However, no language and time restrictions were applied. The details of the search strategy are presented in the Appendix A.

### 2.3. Selection Criteria

We included cross-sectional studies reporting data on the prevalence of primary (treatment-naive) antibiotic resistance in *H. pylori* to the following antibiotics: metronidazole, clarithromycin, tetracycline, amoxicillin, ciprofloxacin, levofloxacin, and furazolidone among the eight South Asian countries (Bangladesh, Bhutan, India, Maldives, Nepal, Pakistan, and Sri Lanka). The duplicate studies or publications reporting data on the same study were used only once by including only the most detailed and recently published data. 

We excluded studies reporting prevalence data on fewer than five *H. pylori* isolates; studies reporting resistance prevalence only as a percentage with no mention of the total number of isolates; review articles, conference papers, poster publications, case reports, and articles with incomplete reporting of data and irretrievable full-text articles. 

### 2.4. Selection Process

The five databases were searched systematically by two authors (PP and SAM). Studies obtained from the electronic databases were exported to Mendeley version 1.19.8 reference software in compatible formats. Duplicate articles were screened first by the reference software and then manually. After that, duplicates were recorded and deleted. The selection process involved two steps: first, the relevant studies were retrieved and evaluated independently by each reviewer. Secondly, the disagreements were discussed and resolved by consensus, thus deciphering the final data.

### 2.5. Definitions

Patients who tested positive by any of the following tests: histology, serology, polymerase chain reaction (PCR), culture, stool antigen, urea breath test, or rapid urease test were considered infected by *H. pylori*. In accordance with international guidelines, a threshold of 15% was adopted as resistance [10,11]. Both phenotypic tests, which included the Epsilometer test (E-test) and agar dilution, and genotypic tests, which included nucleic acid-based tests (PCR) were taken into account for determining the resistance. Resistance developed prior to the initiation of the first eradication treatment was identified as primary resistance. The response to the treatment was analyzed on the basis of noninvasive and invasive (which included culture negative after gastric biopsy) tests. 

### 2.6. Data Extraction

Data were extracted concerning the following: author, publication year, country, sample size, age, sex, patients with underlying disease, healthy/symptomatic patients, antibiotic susceptibility test, and antibiotic resistance to each aforementioned antibiotic. 

### 2.7. Quality Assessment

The quality assessment of the included studies was performed using the Newcastle–Ottawa scale tool adapted for cross-sectional studies [17]. Two authors (PP and SAM) evaluated the studies independently. Each study was provided with a score ranging from 0 to 10. A study with a score ≥7 was considered to have a low risk of bias, a study with a score of 4–6 was considered to have an intermediate risk of bias, and a study with a score ≤4 was considered to have a high risk of bias. Any cases of discrepancies were resolved by the common consensus. Details of quality assessment are in the Appendix A.

### 2.8. Statistical Analysis 

All basic calculations were performed in Microsoft Excel 2016 (Microsoft Corp., Redmond, WA, USA). For further analysis, data from the Excel sheet were extracted using STATA version 17.0 (Stata Corp., College Station, TX, USA). Pooling of primary resistance was performed with Der Simonian and Laird’s random-effects model due to different populations and demographic settings across studies [18]. With this difference, we anticipated considerable heterogeneity among the studies. The Cochrane Q test and I^2^ statistics were used to examine the heterogeneity between studies. Substantial heterogeneity was measured for a value of I^2^ > 75% [19]. All analysis was two-tailed with a significance level set to <0.05. For the determination of the source of heterogeneity, further subgroup analysis was performed for the method of detection of *H. pylori* and resistance (single or combined method), study country, and study period (2003–2012, 2013–2022). Publication bias was evaluated by the funnel plot of the overall effect size with standard error (SE). For the small study effect size, Egger’s regression test was performed. A *p*-value < 0.1 was regarded as statistically significant for publication bias.

## 3. Results

### 3.1. Study Selection and Characteristics

Altogether, 430 articles were obtained from the databases of PubMed, Embase, Cochrane Library, Web of Science, and Google Scholar, from inception to September 2022. From the initial search, 84 duplicate articles were removed. From the remaining 346 articles, 309 articles were removed by screening the title and abstract. A full-text review was performed thoroughly in the remaining 37 articles, out of which 14 articles were excluded based on the eligibility criteria. Finally, 23 studies were included in the analysis. The PRISMA diagram tailoring the details of the study selection process is shown in Figure 1. 

These 23 studies were conducted in five South Asian countries (Bangladesh = 3, Bhutan = 2, India = 11, Nepal = 1, and Pakistan = 7). No studies were reported from the remaining three South Asia countries (Afghanistan, Maldives, and Sri Lanka). A map of South Asia with the included countries for the analysis is depicted in the Appendix A. 

A total of 6357 patients were enrolled, out of which 3294 isolates of *H. pylori* tested positive (51.8%, 95%CI: 0.51–0.53). However, only 2192 samples of the 3249 isolates were taken to record antibiotic resistance (67.5%, 95%CI: 0.66–0.69). With the available data, the total antibiotic susceptibility was calculated to be 74.35% (95%CI: 0.72–0.76), the male-to-female ratio was 1.26:1 and all studies included symptomatic patients i.e., either dyspepsia, reflux disease, peptic ulcer, gastritis or duodenitis and gastric carcinoma, except for Mahant et al. [20], who included both symptomatic and asymptomatic patients.

Using any one of the standard techniques for antibiotic susceptibility as aforementioned in the inclusion criteria, 2192 samples were tested for resistance. Three studies used the dual method for analyzing antibiotic resistance, whereas the rest adopted a single method such as agar dilution, disk diffusion, E-test, and PCR. The detailed characteristics of the included studies are illustrated in Table 1.

### 3.2. Study Quality

Using the Newcastle–Ottawa scale tool for quality assessment of the studies, 4 studies had an intermediate/moderate risk of bias, and the remaining 19 studies had a low risk of bias. There were no studies with a high risk of bias. So, all studies were included in the final meta-analysis. The details of the quality assessment are shown in the Appendix A. 

### 3.3. Pooled Prevalence of Helicobacter pylori Resistance

The pooled prevalences of commonly used antibiotic resistances in South Asia were: amoxicillin 23% (95%CI: 0.15–0.30, I^2^ = 99.80%), clarithromycin 27% (95%CI: 0.17–0.38, I^2^ = 99.76%), metronidazole 69% (95%CI: 0.62–0.76, I^2^ = 99.74%), tetracycline 16% (0.06–0.25, I^2^ = 99.83%), ciprofloxacin 12% (95%CI: 0.04–0.20, I^2^ = 94.52%), levofloxacin 34% (95%CI: 0.22–0.47, I^2^ = 98.69%), and furazolidone 14% (95%CI: 0.06–0.22, I^2^ = 59.81%). Figure 2, Figure 3, Figure 4, Figure 5, Figure 6, Figure 7 and Figure 8 show forest plots of individual antibiotic resistance. 

### 3.4. Subgroup Analysis 

Except for furazolidone, the degree of heterogeneity was substantial in the pooled resistance of antibiotics. So, to further explore the source of heterogeneity, subgroup analysis was conducted for the country, trend analysis of 10-year intervals, method of detection of *H. pylori*, and method of detection of *H. pylori* resistance.

#### 3.4.1. Country-Wise Analysis

Among eight countries in South Asia, data were retrieved for only five countries. Higher prevalences were seen in Pakistan, India, and Bangladesh. The pooled prevalence of common antibiotic resistance to *H. pylori* in different countries is shown in Table 2. Despite this, there was no substantial reduction in the degree of heterogeneity.

#### 3.4.2. Analysis of the Trend

Antibiotic resistance was stratified in 10-year intervals from 2002 to 2012 and 2013 to 2022 to analyze the pattern of resistance over a 20-year period. Resistance to ciprofloxacin, clarithromycin, and tetracycline has increased significantly in the past 20 years. However, the increase in antibiotic resistance was statistically significant only for tetracycline (*p* = 0.04). From the trend analysis, the degree of heterogeneity decreased for ciprofloxacin but did not make substantial changes for other antibiotics. The details of the trend analysis are shown in Table 3.

#### 3.4.3. Method of Detection of *H. pylori* and Antibiotic Resistance Detection Method

Despite the subgroup analysis for *H. pylori* detection and antibiotic resistance detection methods, heterogeneity was considerable for both subgroups. Details are shown in the Appendix A.

### 3.5. Sensitivity Analysis

For metronidazole and levofloxacin, there were no significant changes in the overall results and the degree of heterogeneity when studies were omitted one by one. However, for amoxicillin, clarithromycin, and tetracycline, the study of Siddiqui et al. [15] had significant changes in the overall prevalence. Moreover, for ciprofloxacin, the study of Pandya et al. [21] had a significant effect. The details of the sensitivity analysis are shown in the Appendix A.

### 3.6. Publication Bias 

Funnel plots of levofloxacin, ciprofloxacin, and furazolidone were relatively symmetrical. However, the funnel plots of amoxicillin, clarithromycin, and metronidazole were relatively asymmetrical. Egger’s test for small study effect size showed the presence of a small study effect for ciprofloxacin (*p* = 0.002), and metronidazole (*p* < 0.001). No small study effect was observed for the remaining antibiotics (*p* > 0.1). The details of publication bias are shown in the Appendix A.

## 4. Discussion

The discovery of antibiotics has helped us to treat infectious diseases in a way that we had never imagined. However, many decades down the road, with the emergence of antibiotic resistance due to unjustified use of them, bacterial infections have again become a considerable threat [41]. As far as we are aware, this is the first meta-analysis analyzing the antibiotic resistance patterns of *H. pylori* in South Asia. We have analyzed the *H. pylori* resistance pattern to commonly prescribed antibiotics including clarithromycin, amoxicillin, metronidazole, quinolones including ciprofloxacin, levofloxacin, furazolidone, and tetracycline. 

This study found the prevalence of amoxicillin resistance to be 23% in South Asia, which is higher than in Southeast Asia (2%), the Western Pacific (1%), Europe (0%), the Americas (10%), and the East Mediterranean (14%). However, the resistance is lower compared to Africa (38%) [42]. Clarithromycin resistance was 27% in South Asia, which is comparable to the East Mediterranean (33%), and West Pacific (34%); but is significantly higher than in Africa (15%), the Americas 10%), Southeast Asia (10%) and Europe (18%) [42]. Similarly, in South Asia, metronidazole resistance was 69%, which is much higher than in America (23%), and Europe (32%) [42]. Moreover, the tetracycline (16%), and levofloxacin (34%) resistance in South Asia is significantly higher than in the other regions of the world. A recent meta-analysis showed antibiotic resistance in Asia for *H. pylori*, amoxicillin (3.9%), clarithromycin (30.9%), metronidazole (48.9%), and levofloxacin (29.9%) [43]. In general, antibiotic resistance patterns in South Asia are significantly higher than the resistance patterns in the rest of the world. The global burden of disease (GBD) in 2019 showed that South Asia was only after Africa in terms of deaths attributable to antibiotic resistance [44]. 

The major mode of transmission of *H. pylori* is via contaminated water and food. Combined with poor knowledge and education about hygiene, the transmission of resistant antibiotic strains is a high risk in South Asia. United Nations International Children’s Emergency Fund (UNICEF) has estimated that over 134 million people in South Asia still do not have access to improved drinking water and around 68 to 84 percent of water sources are contaminated [45]. The wide availability of antibiotics without prescription or even inappropriate prescription is common in South Asia. Studies have shown the prevalence of self-medication in South Asian countries to be near around 50 percent [46,47,48]. Poor antibiotic stewardship, poorly enforced food security policies, contaminated food products, poor implementation of infection prevention programs, limited resources, poor awareness among healthcare professionals, and non-adherence to treatment guidelines are also responsible for the higher burden of antibiotic resistance in South Asian countries [49]. 

Among the antibiotics, metronidazole seems to have the highest resistance rate (69%). A recent systematic review observed 26–96.4% resistance to metronidazole in Southeast Asia nations [50]. This is seemingly clear that the widespread use of inexpensive metronidazole in parasitic infections (amebic dysentery), gynecological infections, bone and joint infections, postoperative medication, septicemia, oral infections, and endocarditis. Moreover, it is considered a first-line drug in the Southeast Asia *H. pylori* management regimen [51]. So, we can conclude that the high resistance rate of metronidazole is due to its excessive use and a high degree of non-compliance among patients with treatment regimens [52]. 

Clarithromycin resistance is also high (27%) in South Asia. Point mutations in the 23S ribosomal subunit of *H. pylori*, mainly at positions 2142 and 2143 with a transition of adenine to guanine, are responsible for the resistance to clarithromycin [53]. Additionally, clarithromycin resistance is because of its frequent use in pediatric, respiratory, and otorhinolaryngology-related illnesses [54]. As clarithromycin resistance is more than 15%, clarithromycin containing triple therapy should no longer be the standard treatment modality in South Asia. So, two alternative options, one using bismuth quadruple therapy and another treating based on microbiological tests, should be followed in South Asian countries. 

Amoxicillin is frequently used by the people of the South Asian region in most medical issues. This is due to the over-the-counter availability of the drug, unnecessary prescriptions, and patient preference to take antibiotics for non-infectious illnesses. All these factors have contributed to the higher amoxicillin resistance (23%) even in *H. pylori* infection.

Similarly, the tetracycline groups of drugs are increasingly being used for skin infections and for anaerobes. This has resulted in subsequent resistance among *H. pylori* infection cases. Based on the meta-analysis, tetracycline resistance is higher (17%) compared to the global trends. The studies of Pandya et al. [21] and Siddiqui et al. [15] had a certain degree of outlier effect in the meta-analysis, thus affecting the overall pooled result of tetracycline resistance.

Furazolidone was used in *H. pylori* treatment in India in the studies of Gehlot et al. [25], Pandey et al. [21], and Malhotra et al. [27]. Despite of its infrequent use, the resistance was high (14%). The probable reason for this is because of its easy availability and frequent use for the treatment of cholera, giardiasis, and colitis in the Indian subcontinent.

Among fluroquinolones, levofloxacin resistance (34%) is higher compared to ciprofloxacin (12%). The exact reason for the variable degree of resistance among these quinolones has not been discussed in the literature. However, according to this meta-analysis, the studies of Mahant et al. [20], Aftab et al. [22], and Shetty et al. [34] had a significant outlier effect while computing the pooled resistance for levofloxacin. This can be interpreted as a high rate of levofloxacin resistance in certain regions of India and Bangladesh. The high levofloxacin resistance is probably due to the cross-resistance with high fluoroquinolone resistance in South Asia. Moreover, Mahant et al. observed that the *H. pylori* strains from northeast India form a different cluster based on the partial sequence of 23S rRNA and gyrase A genes, resulting in high fluoroquinolone resistance [20].

From subgroup analysis: amoxicillin, clarithromycin, and tetracycline resistances were highest in Pakistan; whereas ciprofloxacin and levofloxacin resistance was highest in India. Metronidazole resistance was highest in Bhutan (81%) followed by India (80%). These differences in resistance likely reflect the level of urbanization, sanitation, access to clean water, and varied socioeconomic status among South Asian countries [55]. There are significant differences in *H. pylori* prevalence even within the same country. Similar to this, a meta-analysis from Pakistan showed a higher prevalence of resistance to other commonly used antibiotics and a lack of an effective surveillance system and proper antibiotic stewardship [56]. However, due to the overlapping confidence intervals, these results cannot be interpreted as statistically significant.

Ten-year trend analysis showed the increasing resistance prevalence for clarithromycin (21% to 30%), ciprofloxacin (3% to 16%), and tetracycline (5% to 20%) from 2003 to 2022. This can be explained by the increasing use of macrolides, fluoroquinolones, and tetracyclines for respiratory, urinary, and skin infections due to the low cost, easy availability, and low side-effect profiles [13]. Moreover, the consumption of fluoroquinolones has increased by 64%, and of macrolides by 19%, between 2000 and 2010, thus leading to increasing trends of antibiotic resistance [57]. Fluoroquinolone monotherapy as an alternative first-line therapy for community-acquired pneumonia has resulted in increasing rampant consumption at community levels in South Asia [58]. 

Studies have shown an association between *H. pylori* infection and gastric carcinoma. The most commonly accepted hypothesis is that virulence factors such as cytotoxin-associated gene A, vacuolating cytotoxin A, and other types of outer membrane proteins are responsible for gastric cancer [59]. A recent meta-analysis estimated the frequency of gastric cancer in the *H. pylori*-infected population to be 17.4% [60]. Similarly, more than 90% of non-cardia gastric cancers occur due to *H. pylori* infection [61,62]. As the resistance of *H. pylori* to currently used antibiotics is significantly higher in South Asia, this increases the cases of chronic infection, and thus increases the risk of gastric cancer and cancer-related deaths. 

In order to tackle the increasing antibiotic resistance, first and foremost, *H. pylori* infection prevention and control by means of community-based programs focusing on water, sanitation, and hygiene is essential. Second, attempts to reduce the exposure to antibiotics and rampant self-administration of antibiotics should be taken into control by means of effective implementation of antibiotic stewardship programs in South Asian countries [63]. First-line treatment strategies should be adapted according to the resistance patterns of the individual countries. Standard guidelines recommend the use of alternative treatment regimens where the resistance prevalence is more than 20%. Moreover, the inclusion of newer treatment modalities such as the use of vonoprazan [64], plant and animal products, antimicrobial peptides, metallic nanoparticles, liposomal formulations [50], and the introduction of a vaccine against *H. pylori* [65] are extremely important in the eradication of *H. pylori* from South Asia.

This meta-analysis has certain limitations. First, the heterogeneity among enrolled studies was remarkable. This can be explained by the variation in the study period, study countries, and the methods used to measure antibiotic resistance. Despite conducting subgroup analysis, the heterogeneity was still high. Second, no studies were included from Sri Lanka, Maldives, and Afghanistan and there was a single study from Nepal. So, the pooled prevalence might not represent the true burden of antibiotic resistance in South Asia. Third, participants in the included studies were enrolled from medical centers and thus might not be representative of the general population. Fourth, during the period of 2003–2022; the trends of antibiotic resistance were changing and different countries were implementing different antibiotic stewardship plans. These factors might have affected the results of our analysis. Fifth, there was a single study for the years 2021 and 2022, respectively, so the 10-year trend analysis for the time 2013–2022 might have fewer implications. So, readers and clinicians are requested to carefully interpret the findings of our meta-analysis.

## 5. Conclusions

This meta-analysis concludes the high resistance rate of commonly prescribed antibiotics used for the treatment of *H. pylori* infection in South Asia. The highest resistance rate is for metronidazole, followed by levofloxacin and clarithromycin, respectively. As clarithromycin resistance is more than 15 percent in Bangladesh, India, and Pakistan, bismuth quadruple therapy is recommended as the first-line therapy, instead of clarithromycin-containing triple therapy. However, studies assessing clarithromycin resistance in Nepal, Bhutan, and other South Asian countries were limited, so either clarithromycin containing triple therapy or bismuth quadruple therapy can be used in these countries until the exact resistance patterns are studied. Alternatively, antibiotic susceptibility tests can be performed before prescribing antibiotics. 

Furthermore, large-scale community-based studies comparing resistant patterns in naive and treatment failure cases are mandated in these countries. Moreover, a robust surveillance system and strict adherence to antibiotic stewardship are required for minimizing antibiotic resistance in the future. 

## Figures and Tables

**Figure 1 tropicalmed-08-00172-f001:**
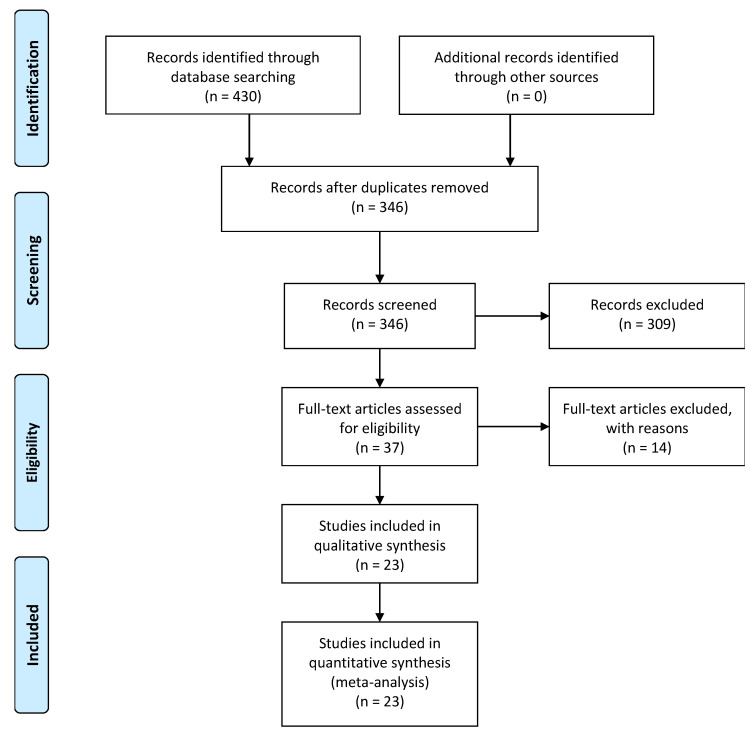
PRISMA Diagram Detailing the Study Selection Process.

**Figure 2 tropicalmed-08-00172-f002:**
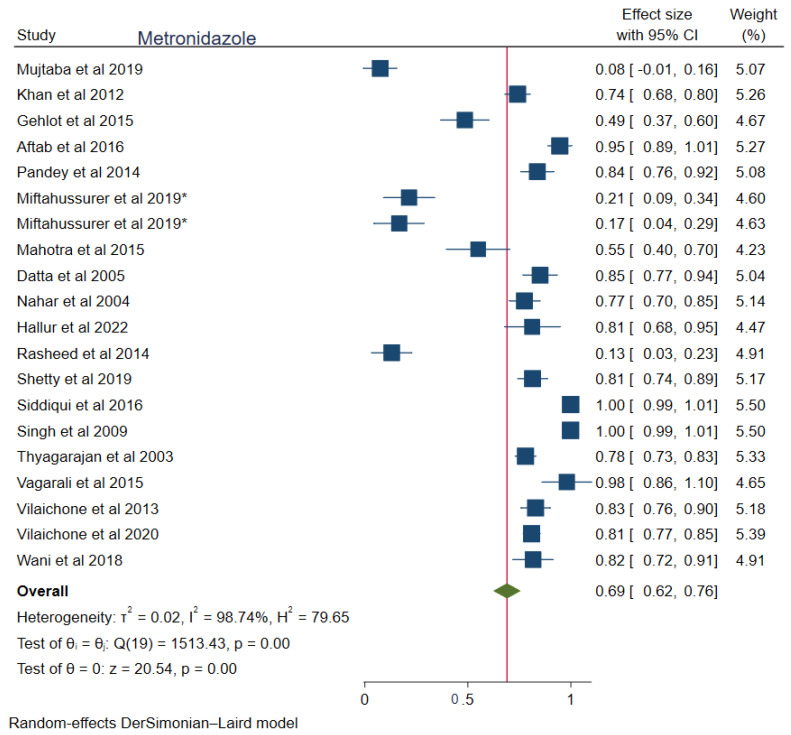
Forest Plot Showing Metronidazole Resistance. (*: the same study with data from two countries, Reference Studies: [14,15,21,22,23,24,25,26,27,29,30,31,33,34,35,36,37,38,39]).

**Figure 3 tropicalmed-08-00172-f003:**
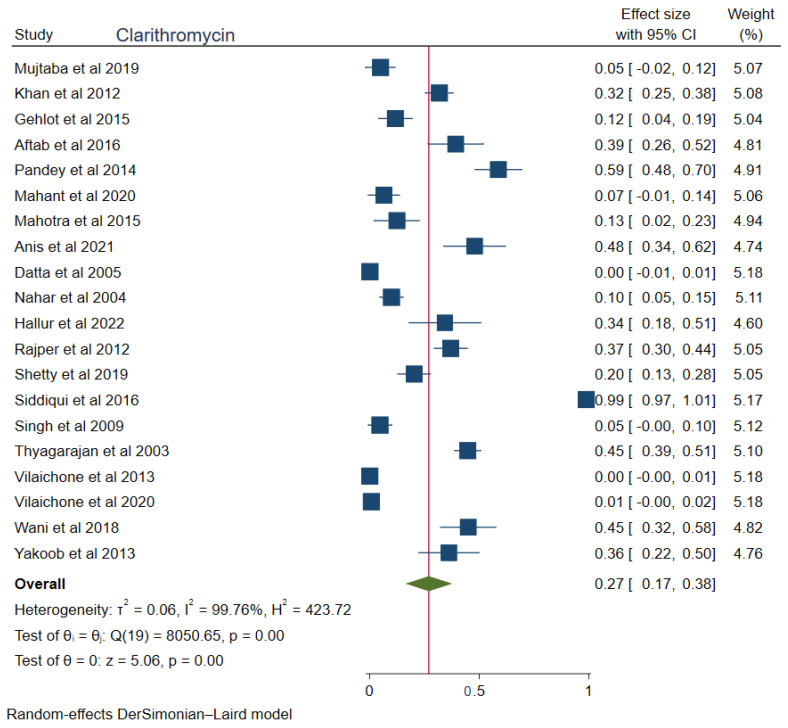
Forest Plot Showing Clarithromycin Resistance. (Reference Studies: [15,20,21,22,23,24,25,27,28,29,30,31,32,34,35,36,37,38,39,40]).

**Figure 4 tropicalmed-08-00172-f004:**
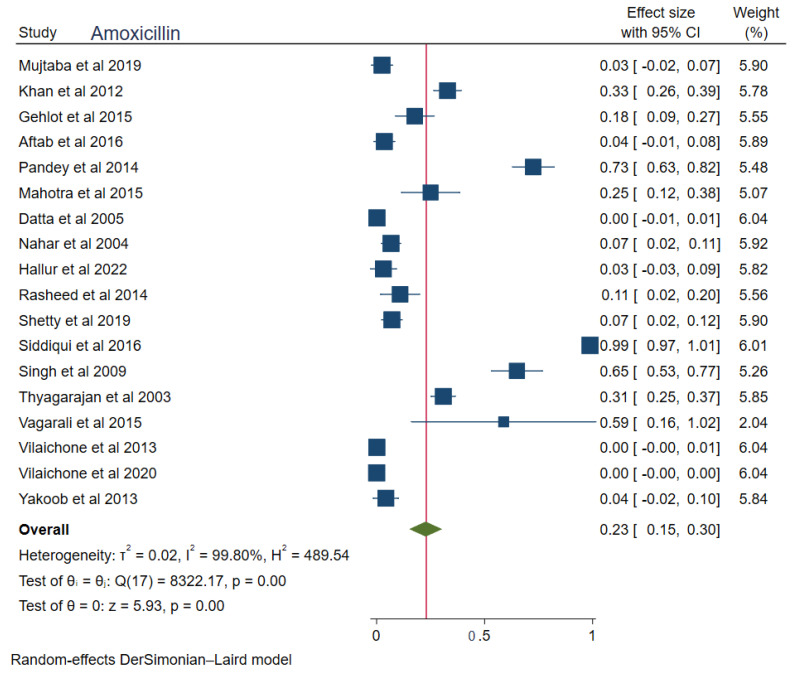
Forest Plot Showing Amoxicillin Resistance. (Reference studies: [15,21,22,23,24,25,27,29,30,31,33,34,35,36,37,38,40]).

**Figure 5 tropicalmed-08-00172-f005:**
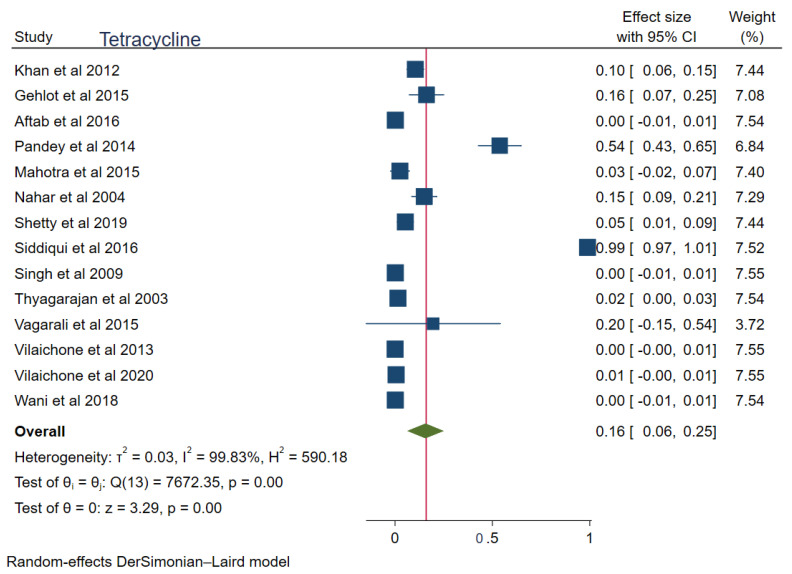
Forest Plot Showing Tetracycline Resistance. (Reference studies: [14,15,21,22,24,25,27,30,34,35,36,37,38,39]).

**Figure 6 tropicalmed-08-00172-f006:**
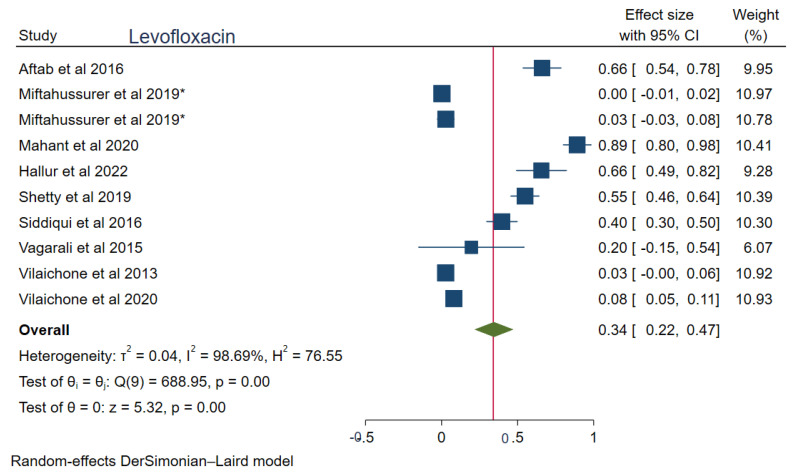
Forest Plot Showing Levofloxacin Resistance. (*: the same study with data from two countries, Reference studies: [14,15,20,22,26,31,34,37,38]).

**Figure 7 tropicalmed-08-00172-f007:**
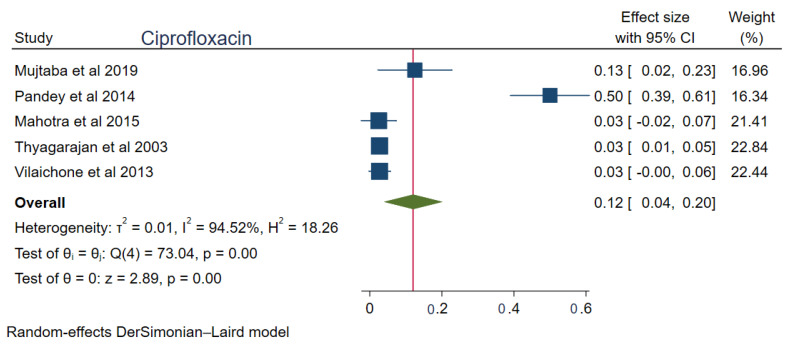
Forest Plot Showing Ciprofloxacin Resistance. (Reference studies: [21,23,27,36,37]).

**Figure 8 tropicalmed-08-00172-f008:**
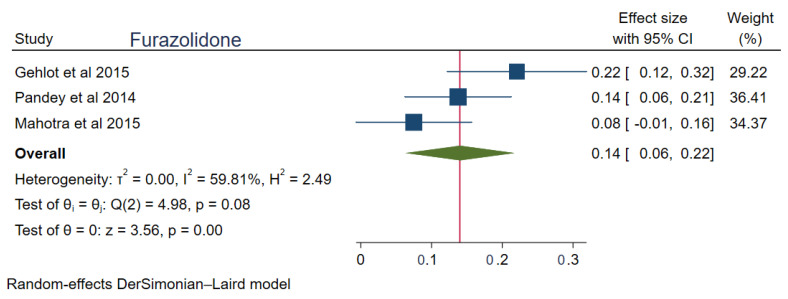
Forest Plot Showing Furazolidone Resistance. (Reference studies: [21,25,27]).

**Table 1 tropicalmed-08-00172-t001:** Characteristics of the included studies.

S.N.	Author	Year	Country	Male	Female	Age in yrs.	Method of *H. pylori* Detection	Method of Antibiotic Susceptibility Detection	Sample for *H. pylori* Resistance
1.	Pandya et al. [21]	2014	India	518	337	10–90	Culture	Disk diffusion	80
2.	Aftab et al. [22]	2016	Bangladesh	26 ^§^	30	35.2 ± 11.8	Culture	Agar dilution	56
3.	Mujtaba et al. [23]	2019	Pakistan	24 ^§^	16	35–55	Culture	E-test	40
4.	Khan et al. [24]	2012	Pakistan	NA	NA	NA	Culture, PCR	PCR	201
5.	Gehlot et al. [25]	2015	India	36 ^§^	32	18–86	Culture	Agar dilution	68
6.	Miftahussurer et al. ^†^ [26]	2019	Nepal	NA	NA	NA	Culture	Agar dilution	42
7.	Miftahussurer et al. ^†^ [26]	2019	Bangladesh	NA	NA	NA	Culture	Agar dilution	36
8.	Mahant et al. [20]	2020	India	32 ^§^	14	18–80	Culture	Agar diffusion, PCR	46
9.	Malhotra et al. [27]	2015	India	39	11	15–50	Culture	Disk diffusion	40
10.	Anis et al. [28]	2021	Pakistan	NA	NA	NA	Culture, rapid urease test	Agar dilution	48
11.	Datta et al. [29]	2005	India	72	31	21–71	Culture	Agar dilution	67
12.	Nahar et al. [30]	2004	Bangladesh	202	76	15–78	Culture	Agar dilution	120
13.	Hallur et al. [31]	2022	India	85	28	46.85 ± 13.2	Culture, PCR	E-test	32
14.	Rajper et al. [32]	2012	Pakistan	84 ^§^	78	9–75	H&E	PCR	162
15.	Rasheed et al. [33]	2014	Pakistan	71	22	46 ± 16.4 for males 49.1 ± 15.1 for females	H&E	PCR	46
16.	Shetty et al. [34]	2019	India	80 ^§^	33	46.2 ± 14	H&E	PCR	113
17.	Siddiqui et al. [15]	2016	Pakistan	424	465	35.6 ± 14.32	Gram staining and urease test	Disk diffusion	93
18.	Singh et al. [35]	2009	India	68	40	18–75	Gram staining and urease test and PCR	Agar dilution and E-test method	63
19.	Thyagarajan et al. [36]	2003	India	NA	NA	NA	H&E and urease, catalase test	Disk diffusion, and E-test	259
20.	Vagarali et al. [14]	2015	India	152	48	NA	Culture, H&E, and urease test	Disk diffusion	5
21.	Vilaichone et al. [37]	2013	Bhutan	51 ^§^	60	36.8 ± 13.9	Culture	E-test	111
22.	Vilaichone et al. [38]	2020	Bhutan	514	664	41.5 ± 15.2	Histopathology, urease, and culture	E-test	357
23.	Wani et al. [39]	2018	India	54	41	46.78	Histopathology, urease, and culture	PCR	60
24.	Yakoob et al. [40]	2013	Pakistan	80	40	41 ± 13	Histopathology, urease, C14 urea breath test, and culture	E-test	47

^†^: the same study with data from two countries, ^§^: from *H. pylori*-positive sample, PCR: polymerase chain reaction, E-test: Epsilometer test, H&E: hematoxylin and eosin staining, NA: not available.

**Table 2 tropicalmed-08-00172-t002:** Country-wise Antibiotic Resistance Patterns.

A. Amoxicillin (Pooled prevalence: 23%)	
Bangladesh	5% (95%CI: 0.02–0.09); *p* = 0.36
India	29% (95%CI: 0.15–0.44); *p* < 0.001
Pakistan	30% (95%CI: −0.21–0.81); *p* < 0.001
B. Ciprofloxacin (Pooled prevalence: 12%)	
Bhutan	3% (95%CI: 0.00–0.06); *p* < 0.001
India	17% (95%CI: 0.01–0.34); *p* < 0.001
Pakistan	13% (95%CI: 0.02–0.23); *p* < 0.001
C. Clarithromycin (Pooled prevalence: 27%)	
Bangladesh	24% (95%CI: 0.05–0.53); *p* < 0.001
India	23% (95%CI: 0.11–0.36); *p* < 0.001
Pakistan	43% (0.03–0.82); *p* < 0.001
D. Metronidazole (Pooled prevalence: 69%)	
Bangladesh	63% (95%CI: 0.26–1.00); *p* < 0.001
Bhutan	81% (95%CI: 0.78–0.85); *p* = 0.64
India	80% ((95%CI: 0.7–0.9); *p* < 0.001
Nepal	21% (95%CI: 0.09–0.34); *p* < 0.001
Pakistan	49% (95%CI: 0.03–0.94); *p* < 0.001
E. Tetracycline (Pooled prevalence: 16%)	
Bangladesh	7% (95%CI: –0.07–0.22); *p* < 0.001
India	7% (95%CI: 0.03–0.10); *p* < 0.001
Pakistan	55% (95%CI: –0.32–1.41); *p* < 0.001
F. Levofloxacin (Pooled prevalence: 34%)	
Bangladesh	34% (95%CI: –0.28–0.96); *p* < 0.001
Bhutan	5% (95%CI: 0.00–0.11); *p* = 0.01
India	61% (95%CI: 0.38–0.84); *p* < 0.001
Pakistan	40% (95%CI: 0.30–0.50); *p* < 0.001
G. Furazolidone (Pooled prevalence: 14%)	
India	14% (95%CI: 0.06–0.22); *p* < 0.001

*p* < 0.05 is considered statistically significant.

**Table 3 tropicalmed-08-00172-t003:** Analysis of antibiotic resistance in 10-year intervals from 2003 to 2022.

Antibiotic	2003–2012 (95%CI)	2013–2022 (95%CI)	*p*-Value
Ciprofloxacin	3% (0.01–0.05)	16% (0.02–0.30)	0.28
Clarithromycin	21% (0.06–0.37)	30% (0.14–0.46)	0.25
Tetracycline	5% (0.02–0.09)	20% (0.07–0.27)	0.04
Metronidazole	83% (0.69–0.97)	63% (0.48–0.78)	0.82
Amoxicillin	26% (0.09–0.44)	22% (0.12–0.32)	0.9

## Data Availability

This section provides details regarding where data supporting reported results can be found, including links to publicly archived datasets analyzed or generated during the study.

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
