# Peer review of "Drug Resistance Patterns of Commonly Used Antibiotics for the Treatment of Helicobacter pylori Infection among South Asian Countries: A Systematic Review and Meta-Analysis"

_tropicalmed, 2023, doi:10.3390/tropicalmed8030172_

Round 1

Reviewer 1 Report

The current article (ropicalmed-2217518) highlighted the increasing trend of antibiotic resistance in H.pylori in South Asian countries based on systematic review of literature. The theme and findings of study will be of great interest for readers and scientific community to know the pattern of drug resistance in H. pylori.

The following comments should be properly addressed.

Comments:

  • There are numerous grammatical and formatting mistakes which should be rectified,
  •  The importance of the study should be revised in the abstract section.
  • Page 2, line 60: gram-negative may be changed to Gram-negative
  •  Page 2, line 65: The scientific name should be in italic form, such as Helicobacter pylori
  • Helicobacter pylori once defined should be written as H. pylori later on.
  • Page 2, line 66: The sentence (the prevalence of H. pylori infection in South 66 Asia is 57.7%, which is higher than the prevalence in Asia (44.7%)) should be revise scientifically.
  • Table 1: Formatting of table 1 required. Total resistance heading in the last column is scientifically wrong. It must revised and must truly represent the contents in the column.
  • The abbreviations used in tables should be defined in the footnotes.
  • Table 2: For some antibiotics resistance pattern was explored in three countries (Bangladesh, India and Pakistan) while in some five countries (Bangladesh, India and Pakistan, Bhutan, Nepal). So same pattern may be followed for better understanding of readers. P-value should be defined in the foot note.
  • There are many studies increasing antibiotic resistance in H. pylori from Iran (A South Asian country). Why the data from Iran has not been presented in the article?
  • Table 3: It is not scientifically sound to represent antibiotic resistance based on only one study from 2022. So the range may be revised 2020-2021.
  • The limitation of the study should be properly described,

Author Response

S.N.

Comment

Answer

1.      

There are numerous grammatical and formatting mistakes which should be rectified.

Thank you for the comment.

We have revised the grammatical and formatting mistakes in this version of the manuscript.

2.      

The importance of the study should be revised in the abstract section.

Thank you for the comment.

Accordingly, we have revised the importance of study in abstract section as follows:

Background: In South Asia, resistance to commonly used antibiotics for the treatment of Helicobacter pylori infection are increasing. Despite of this, accurate estimates of overall antibiotic resistance are missing. Thus, this review aims to analyze the resistance rates of commonly used antibiotics for the treatment of H. pylori in South Asia.

3.      

Page 2, line 60: gram-negative may be changed to Gram-negative.

Thank you for the comment.

Accordingly, we have made the necessary changes.

4.      

 Page 2, line 65: The scientific name should be in italic form, such as Helicobacter pylori.

Thank you for the comment.

Accordingly, we have revised the line.

5.      

Helicobacter pylori once defined should be written as H. pylori later on.

Thank you for the comment.

Considering your valuable comment, we have revised all the H. pylori in Italics.

6.      

Page 2, line 66: The sentence (the prevalence of H. pylori infection in South 66 Asia is 57.7%, which is higher than the prevalence in Asia (44.7%)) should be revise scientifically.

Thank you for the comment.

Accordingly, we have revised the line.

7.      

Table 1: Formatting of table 1 required. Total resistance heading in the last column is scientifically wrong. It must be revised and must truly represent the contents in the column.

Thank you for the comment.

Accordingly, we have revised the Table 1 and made it more accurate, scientific and easy to understand.

8.      

The abbreviations used in tables should be defined in the footnotes.

Thank you for the comment.

Accordingly, we have defined the abbreviations in the footnotes of the respective tables.

9.      

Table 2: For some antibiotics resistance pattern was explored in three countries (Bangladesh, India and Pakistan) while in some five countries (Bangladesh, India and Pakistan, Bhutan, Nepal). So same pattern may be followed for better understanding of readers. P-value should be defined in the foot note.

Thank you for the comment.

Based on the common consensus, we had an aim to explore the prevalences of the individual countries. But drug resistance for some antibiotics was not available from some countries. So, in order to make a more inclusive meta-analysis we are unable to remove the prevalence data from studies where 5 counties were included. Furthermore, other reviewers have suggested to make changes in the conclusion based on the countrywise antibiotic resistances. We request you to understand our situation.

Additionally, we have mentioned the respective p values in the table, and described it in the footnote.

10.   

There are many studies increasing antibiotic resistance in H. pylori from Iran (A South Asian country). Why the data from Iran has not been presented in the article?

Thank you for the comment.

According to the World Bank, and World Health Organization South Asia is comprised of eight countries namely: Afghanistan, Bangladesh, Bhutan, India, Maldives, Nepal, Pakistan, and Sri Lanka. As, we have followed aforementioned reports, Iran was not included in this study.

11.   

Table 3: It is not scientifically sound to represent antibiotic resistance based on only one study from 2022. So the range may be revised 2020-2021.

Thank you for the comment.

If we did subgroup analysis till 2021, the comparison will be odd. i.e. 10 years trend (from 2003-2012) vs 9 year trend (from 2013-2021).

Similarly, there is also a single study from the year 2021 as well, which will have less weight to the trend analysis if the study from 2022 is removed. So, we had to add the study from 2022 as well. However, we have added the point raised by you in the limitation of the study as follows:

Fifth, there were a single study from the year 2021 and 2022 respectively, so the 10-year trend analysis for the year 2013-2022 might have less implications.

We request you to accept our situation.

12.   

The limitation of the study should be properly described.

Thank you for the comment.

Accordingly, we have revised the limitations of the study as follows:

This meta-analysis has certain limitations. First, the heterogeneity among enrolled studies was remarkable. This can be explained by variation in the study period, study countries, and the methods used to measure antibiotic resistance. Despite conducting subgroup analysis, the heterogeneity was still high.  Second, no studies were included from Sri Lanka, Maldives, and Afghanistan and there was a single study from Nepal. So, the pooled prevalence might not represent the true burden of antibiotic resistance in South Asia. Third, participants in the included studies were enrolled from medical centres and thus might not be representative of the general population. Fourth, during the period of 2003-2022; the trends of antibiotic resistance were changing and different countries were implementing different antibiotic stewardship plans. These factors might have affected the results of our analysis. Fifth, there were a single study from the year 2021 and 2022 respectively, so the 10-year trend analysis for the time 2013-2022 might have less implications. So, the readers and clinicians are requested to carefully interpret the findings of our meta-analysis.

Reviewer 2 Report

This meta-analysis systematically summarizes the drug resistance information of people infected with Helicobacter pylori in South Asia, which is of reference value to the formulation of prevention and control strategies for Helicobacter pylori infection and the formulation of eradication treatment plans for the corresponding population, but it still needs to be appropriately modified.

1. The sum should be added in each table. For example, Table 2 suggests adding the relevant content of The pooled prevalence of commonly used antibiotic resistances in the text.

2. Because the drug resistance rate of MNZ in Nepal population is only 21%, while the judgment threshold of high drug resistance rate in the consensus is 40% for MNZ. " Our meta-analysis concludes the high resistance rate of commonly prescribed antibiotics used for the treatment of H. pylori infection in South Asia. The highest resistance rate is for metronidazole, followed by levofloxacin, and clarithromycin, respectively. As the clarithromycin resistance is more than 15 percent in this region, bismuth quadruple therapy is recommended as the first line therapy, instead of clarithromycin containing triple therapy." in the conclusion is not suitable for Nepal population. It is suggested  to  treat them differently.

Author Response

S.N.

Comment

Answer

1.

The sum should be added in each table. For example, Table 2 suggests adding the relevant content of “The pooled prevalence of commonly used antibiotic resistances” in the text.

Thank you for the comment.

Accordingly, we have revised the tables.

2.

Because the drug resistance rate of MNZ in Nepal population is only 21%, while the judgment threshold of high drug resistance rate in the consensus is 40% for MNZ. " Our meta-analysis concludes the high resistance rate of commonly prescribed antibiotics used for the treatment of H. pylori infection in South Asia. The highest resistance rate is for metronidazole, followed by levofloxacin, and clarithromycin, respectively. As the clarithromycin resistance is more than 15 percent in this region, bismuth quadruple therapy is recommended as the first line therapy, instead of clarithromycin containing triple therapy." in the conclusion is not suitable for Nepal population. It is suggested to treat them differently.

Thank you for the comment.

Accordingly, we have revised the conclusion as follows:

This meta-analysis concludes the high resistance rate of commonly prescribed antibiotics used for the treatment of H. pyloriinfection in South Asia. The highest resistance rate is for metronidazole, followed by levofloxacin, and clarithromycin, respectively. As the clarithromycin resistance is more than 15 percent in Bangladesh, India, and Pakistan bismuth quadruple therapy is recommended as the first line therapy, instead of clarithromycin containing triple therapy. However, studies assessing clarithromycin resistance from Nepal, Bhutan and other South Asian countries were limited, so either clarithromycin containing triple therapy or bismuth quadruple therapy can be used in these countries until the exact resistance patterns are studied. Alternately, antibiotic susceptibility tests can be performed before prescribing antibiotics.

Furthermore, large-scale community-based studies comparing resistant patterns in naive and treatment failure cases are mandated in these countries. Moreover, a robust surveillance system and strict adherence to antibiotic stewardship are required for minimizing antibiotic resistance in the future.

Reviewer 3 Report

The review entitled “Drug Resistance Patterns of Commonly Used Antibiotics for the Treatment of Helicobacter pylori Infection Among South Asian Countries: A Systematic Review and Meta-Analysis” has an interesting approach. The aim of the study was to evaluate the antibiotic resistance and prevalence of H. pylori to commonly prescribed in South Asia region. The topic is certainly of interest and the literature overview provided is indeed very useful for readers and Health regulatory institutions, but the manuscript needs revision. There are some points that could be included/modified to further improve the quality of the manuscript.

 - The introduction section is very general and should be written more concisely with more focus on the topic of H. pylori prevalence and antibiotic resistance. The authors have mentioned the prevalence of H. pylori in South Asia, but the prevalence in the studied countries has not been mentioned.

- Line 201: “74.3.5%” please correct.

Table 1: The results should be presented following a chronological order, to perceive more clearly the advances.

- Line 222. The title should be in italics.

Line 277-278 and elsewhere: “H. pylori” in italic form.

Lines 284-289: References are needed.

Author Response

S.N.

Comment

Answer

1.      

The introduction section is very general and should be written more concisely with more focus on the topic of H. pylori prevalence and antibiotic resistance. The authors have mentioned the prevalence of H. pylori in South Asia, but the prevalence in the studied countries has not been mentioned.

Thank you for the comment.

Accordingly, we have revised the introduction as follows:

Globally, resistance rates are variable for the antibiotics used for the treatment of H. pylori infection, ranging from 15% to 50% [8]. However, the resistance patterns are constantly high in Asian countries [12].  South Asia comprising of eight countries and one fourth of the global population has high prevalence of antibiotic resistance due to high self-medication rate, poor antibiotic stewardship plans, and changing geopolitical landscape [13]. This has led to high antibiotic resistance in H. pyloriranging upto 98% [14,15].

2.      

Line 201: “74.3.5%” please correct.

Thank you for the comment.

We have revised the line.

3.      

Table 1: The results should be presented following a chronological order, to perceive more clearly the advances.

Thank you for the comment.

Accordingly, we have revised the Table 1.

4.      

Line 222. The title should be in italics.

Thank you for the comment.

Accordingly, we have made the title into italics.

5.      

Line 277-278 and elsewhere: “H. pylori” in italic form.

Thank you for the comment.

Accordingly, H. pylori has been changed to H. pylori (in Italics) everywhere in the manuscript.

6.      

Lines 284-289: References are needed.

Thank you for the comment.

We have added the references in these lines.

Reviewer 4 Report

Drug resistance patterns of commonly used antibiotics for  the treatment of Helicobacter pylori infection is an important topic in the context of increasing of resistance to antibiotic.

I congratulate the authors for the choice of the subject, the well-applied methodology that led to a very good result.

My observations are the follows:

1. compliance with the instructions for authors, including references in the text

2. the name Helicobacter pylori must be written in italics everywhere in the text

3. can table 1 be the additional tab?

4. I think it would be appropriate to mention alternative treatments such as

Cardos et al. Revisiting therapeutic strategies for H. pylori treatment in the context of antibiotic resistance: focus on alternative and complementary therapies

Author Response

S.N.

Comment

Answer

1.

1. compliance with the instructions for authors, including references in the text

Thank you for the comment.

Accordingly, we have revised the manuscript.

2.

the name Helicobacter pylori must be written in italics everywhere in the text

Thank you for the comment.

Accordingly, Helicobacter pylori has been written in Italics everywhere in the manuscript.

3.

can table 1 be the additional tab?

Thank you for the comment.

We think Table 1 comprises a lot of essential information of the meta-analysis. So, we believe it should be in the main manuscript.

We hope you will consider our situation.

4.

 I think it would be appropriate to mention alternative treatments such as

Cardos et al. Revisiting therapeutic strategies for H. pylori treatment in the context of antibiotic resistance: focus on alternative and complementary therapies.

Thank you for the comment.

Accordingly, we have added the alternative treatment strategies suggested by you as follows:

Moreover, the inclusion of newer treatment modalities such as use of vonoprazan [64], plant and animal products, antimicrobial peptides, metallic nanoparticles, liposomal formulations [50], and introduction of vaccine against H. pylori [65] are extremely important in the eradication of H. pylori from South Asia.

Reviewer 5 Report

This systematic review provides a thorough overview of the antimicrobial resistance of Helicobacter pylori to commonly used drugs among South Asian countries. The tables throughout the manuscript are very helpful, and the text does a good job of covering the vast array of this pathogen with major public health importance. It is very helpful to have these approaches summarized in a single paper, and having a short overview of the mechanisms provides a better understanding of each approach. I support its possible publication after appropriate modifications as outlined below:

L60: gram-negative – sentence case

L61: when you cite references please use rectangular brackets intead of normal

L65: Helicobacter pylori – italic (!) please carefully revise the italic writing of the scientific name of all species !!!

L92: The authors must improve the Introduction part of the manuscript emphasizing that the emergence of antibiotic-resistant H. pylori strains has become a global issue, consulting and citing some representative recently articles (e.g. https://doi.org/10.3390/foods11131890, DOI: 10.1097/QCO.0000000000000396 or doi: 10.1089/mdr.2017.0292)

L97: objective of our study” – please avoid the using of personal mode verb formulations. It is not so characteristic for the scientific style. Please revise this concern throughout the manuscript.

L98: “was to evaluate” instead of “is to evaluate”

L123: the antibiotics name must write with lowercase

L214: please carefully revise and format all of the tables according to the journal requirement

L406: the authors must highlight the study limitations within the conclusion section

L442: the reference list is not in agreement with the journal requirement! Please carefully revise this!

Author Response

S.N.

Comment

Answer

1.      

L60: gram-negative – sentence case

Thank you for the comment.

Accordingly, we have revised the issue.

2.      

L61: when you cite references, please use rectangular brackets instead of normal

Thank you for the comment.

Accordingly, we have revised the end text references from simple to squared brackets.

3.      

L65: Helicobacter pylori – italic (!) please carefully revise the italic writing of the scientific name of all species !!!

Thank you for the comment.

Accordingly, we have revised the Helicobacter pylori into Italics in the entire manuscript.

4.      

L92: The authors must improve the Introduction part of the manuscript emphasizing that the emergence of antibiotic-resistant H. pylori strains has become a global issue, consulting and citing some representative recently articles (e.g. https://doi.org/10.3390/foods11131890, DOI: 10.1097/QCO.0000000000000396 or doi: 10.1089/mdr.2017.0292).

Thank you for the comment.

Accordingly, we have added the valid points raised by you in the introduction of the manuscript as follows:

Globally, resistance rates are variable for the antibiotics used for the treatment of H. pylori infection, ranging from 15% to 50% [8]. However, the resistance patterns are constantly high in Asian countries [12].  South Asia comprising of eight countries and one fourth of the global population has high prevalence of antibiotic resistance due to high self-medication rate, poor antibiotic stewardship plans, and changing geopolitical landscape [13]. This has led to high antibiotic resistance in H. pyloriranging upto 98% [14,15]. Proper understanding of the antibiotic resistance pattern in H. pylori through timely and systematic analysis of the available regional data thus is a promising strategy to eliminate H. pylori borne infection and its long-term consequences.

5.      

L97: “objective of our study” – please avoid the using of personal mode verb formulations. It is not so characteristic for the scientific style. Please revise this concern throughout the manuscript.

Thank you for the comment.

Accordingly, we have avoided the personal verbs from the manuscript.

6.      

L98: “was to evaluate” instead of “is to evaluate”

Thank you for the comment.

Accordingly, we have revised.

7.      

L123: the antibiotics name must write with lowercase

Thank you for the comment.

Accordingly, we have revised the manuscript.

8.      

L214: please carefully revise and format all of the tables according to the journal requirement

Thank you for the comment.

Accordingly, we have revised and formatted the table as per the journal requirements.

9.      

L406: the authors must highlight the study limitations within the conclusion section.

Thank you for the comment.

Accordingly, we have revised the conclusion as follows:

This meta-analysis concludes the high resistance rate of commonly prescribed antibiotics used for the treatment of H. pyloriinfection in South Asia. The highest resistance rate is for metronidazole, followed by levofloxacin, and clarithromycin, respectively. As the clarithromycin resistance is more than 15 percent in Bangladesh, India, and Pakistan bismuth quadruple therapy is recommended as the first line therapy, instead of clarithromycin containing triple therapy. However, studies assessing clarithromycin resistance from Nepal, Bhutan and other South Asian countries were limited, so either clarithromycin containing triple therapy or bismuth quadruple therapy can be used in these countries until the exact resistance patterns are studied. Alternately, antibiotic susceptibility tests can be performed before prescribing antibiotics.

Furthermore, large-scale community-based studies comparing resistant patterns in naive and treatment failure cases are mandated in these countries. Moreover, a robust surveillance system and strict adherence to antibiotic stewardship are required for minimizing antibiotic resistance in the future.

10.   

L442: the reference list is not in agreement with the journal requirement! Please carefully revise this!

Thank you for the comment.

Accordingly, we have revised the reference list as per the journal requirements.
